# MRI-Based Machine Learning for Prediction of Clinical Outcomes in Primary Central Nervous System Lymphoma

**DOI:** 10.3390/life14101290

**Published:** 2024-10-11

**Authors:** Ching-Chung Ko, Yan-Lin Liu, Kuo-Chuan Hung, Cheng-Chun Yang, Sher-Wei Lim, Lee-Ren Yeh, Jeon-Hor Chen, Min-Ying Su

**Affiliations:** 1Department of Medical Imaging, Chi Mei Medical Center, Tainan 71004, Taiwan; vince7978@hotmail.com; 2Department of Health and Nutrition, Chia Nan University of Pharmacy and Science, Tainan 71710, Taiwan; 3School of Medicine, College of Medicine, National Sun Yat-Sen University, Kaohsiung 80424, Taiwan; 4Department of Radiological Sciences, University of California, Irvine, CA 92697, USA; yalin0917@gmail.com (Y.-L.L.); jeonhc01@gmail.com (J.-H.C.); msu@uci.edu (M.-Y.S.); 5Department of Anesthesiology, Chi Mei Medical Center, Tainan 710, Taiwan; ed102605@gmail.com; 6Department of Hospital and Health Care Administration, College of Recreation and Health Management, Chia Nan University of Pharmacy and Science, Tainan 71710, Taiwan; 7Department of Neurosurgery, Chi Mei Medical Center, Chiali, Tainan 722, Taiwan; slsw0219@gmail.com; 8Department of Nursing, Min-Hwei College of Health Care Management, Tainan 736, Taiwan; 9Department of Radiology, E-DA Hospital, I-Shou University, Kaohsiung 824, Taiwan; ed103639@edah.org.tw; 10Department of Medical Imaging and Radiological Sciences, College of Medicine, I-Shou University, Kaohsiung 824, Taiwan; 11School of Medicine, College of Medicine, I-Shou University, Kaohsiung 824, Taiwan

**Keywords:** CNS lymphoma, MRI, machine learning, relapse, recurrence

## Abstract

A portion of individuals diagnosed with primary central nervous system lymphomas (PCNSL) may experience early relapse or refractory (R/R) disease following treatment. This research explored the potential of MRI-based radiomics in forecasting R/R cases in PCNSL. Forty-six patients with pathologically confirmed PCNSL diagnosed between January 2008 and December 2020 were included in this study. Only patients who underwent pretreatment brain MRIs and complete postoperative follow-up MRIs were included. Pretreatment contrast-enhanced T1WI, T2WI, and T2 FLAIR imaging were analyzed. A total of 107 radiomic features, including 14 shape-based, 18 first-order statistical, and 75 texture features, were extracted from each sequence. Predictive models were then built using five different machine learning algorithms to predict R/R in PCNSL. Of the included 46 PCNSL patients, 20 (20/46, 43.5%) patients were found to have R/R. In the R/R group, the median scores in predictive models such as support vector machine, k-nearest neighbors, linear discriminant analysis, naïve Bayes, and decision trees were significantly higher, while the apparent diffusion coefficient values were notably lower compared to those without R/R (*p* < 0.05). The support vector machine model exhibited the highest performance, achieving an overall prediction accuracy of 83%, a precision rate of 80%, and an AUC of 0.78. Additionally, when analyzing tumor progression, patients with elevated support vector machine and naïve Bayes scores demonstrated a significantly reduced progression-free survival (*p* < 0.05). These findings suggest that preoperative MRI-based radiomics may provide critical insights for treatment strategies in PCNSL.

## 1. Introduction

Primary central nervous system lymphoma (PCNSL) represents a small fraction, approximately 1–2%, of non-Hodgkin’s lymphoma (NHL) cases and constitutes about 3–4% of tumors found in the central nervous system (CNS) [1]. The predominant histological type of PCNSL is diffuse large B-cell lymphoma (DLBCL) [2]. Currently, there is no universally accepted salvage treatment for PCNSL. The primary treatment approach involves chemotherapy based on methotrexate (MTX), which serves as the foundation of therapy for this disease. Additionally, the use of adjuvant whole-brain radiotherapy (WBRT) following chemotherapy has been shown to improve survival rates compared to chemotherapy alone [3,4]. While high-dose chemotherapeutic agent MTX and radiation therapy that targets the entire brain are recognized as effective treatments for PCNSL, approximately 50% of patients may still experience relapsed or refractory (R/R) disease [5,6]. Various conventional magnetic resonance imaging (MRI) characteristics, including tumor size, non-enhancing hyperintense lesions on T2-weighted images, infratentorial location, and apparent diffusion coefficient (ADC) values, have been identified as significant factors influencing the prognosis of PCNSL [7,8,9]. However, much of the existing data are presented in qualitative and subjective formats, which can lead to inter-rater variability during the interpretation of imaging.

Radiomics analysis is a novel method for imaging diagnosis and prognosis prediction in cancer patients [10]. The method involves extracting a wide range of quantitative features from medical images and analyzing them through various machine learning (ML) algorithms [11]. While several studies have documented a radiomic approach for differentiating between PCNSL and other brain tumors [12,13], there is still a scarcity of radiomic models aimed at predicting clinical outcomes in PCNSL [14,15].

This retrospective study used MRI radiomics analysis to investigate the role of clinical and MRI data in predicting the clinical outcomes of PCNSL. The results offered objective information for developing treatment strategies in PCNSL.

## 2. Related Work

So far, radiomic studies focused on predicting prognosis in PCNSL remain quite rare. Due to the low incidence of PCNSL, most related studies were retrospective, included small datasets, and lacked external validation (Table 1). Furthermore, only one or two MRI sequences were analyzed in the published literature. The comparison between radiomic-based ML and advanced quantitative MRI techniques, such as ADC, for predicting prognosis in PCNSL has not yet been reported.

Among these studies, Nenning et al. [16] reported a relatively large sample size of 191 PCNSL cases across nine institutions in Austria, along with external validation that supports radiomics as a robust predictor of survival. Furthermore, She et al. [17] were the first to use deep learning to predict overall survival (OS) in patients with PCNSL. It is known that the imaging protocols and MRI scanners can significantly influence the extracted features, particularly in a small dataset where it is difficult to select features that are directly related to PCNSL. The strengths and weaknesses of relevant studies were summarized in Table 1.

The five different ML classifiers in our study yielded relatively consistent area under the curve (AUC) and accuracy across all models. This consistency suggests that the features we selected are usable in this dataset. In the future, prospective studies with large datasets and more advanced MRI sequences will be needed to validate the generalizability of radiomic-based ML for predicting clinical outcomes in PCNSL.

## 3. Materials and Methods

### 3.1. Ethics Statement

This study was approved by our institutional review board (Approval No. 10902-009). Since this study was retrospective and did not affect the participants’ healthcare, the requirement for written informed consent was waived. To maintain confidentiality, all patient data were anonymized and de-identified prior to analysis.

### 3.2. Patient Selection

A cohort of 46 consecutive patients, comprising 20 males and 26 females with a median age of 64 years, was selected from Chi Mei Medical Center and E-Da Hospital between October 2005 and November 2020. Inclusion criteria consisted of postoperative histopathological confirmation of PCNSL, availability of complete pretreatment brain MRI and post-treatment brain MRI follow-up, absence of systemic involvement as determined by bone marrow biopsy and PET/CT scans of the thoracic, abdominal, and pelvic regions, and no evidence of immunocompromised diseases or human immunodeficiency virus infection. All the patients were histopathologically diagnosed with DLBCL and received consistent MTX-based induction chemotherapy. The clinical data of the patients, including imaging, medical history, histopathological diagnosis, response, treatments, and survival, were reviewed.

### 3.3. Treatments

First-line therapy for all patients included high-dose MTX (8 g/m^2^/day 1) regimens. During the induction phase of treatment, all patients received high-dose MTX chemotherapy for a duration of 4 to 6 cycles. Alongside MTX, 18 individuals received rituximab (375 mg/m^2^ on day 1) as part of their induction therapy. Dexamethasone was administered to manage neurological symptoms, and intravenous leucovorin was provided as needed. For those who received autologous stem cell transplantation, thiotepa-based conditioning regimens were used for consolidation therapy. WBRT at a dose of 4500 cGy was employed as a salvage treatment for patients with R/R PCNSL. Treatment responses were evaluated according to the criteria established by the International PCNSL Collaborative Group [19]. Of the 46 patients analyzed, 20 were administered consolidation therapy with WBRT, while 2 patients received both WBRT and autologous stem cell transplantation as a result of relapsed or refractory disease.

### 3.4. Evaluation of Response

The response was evaluated by a neuroradiologist (C.C.K., 12 years of neuroradiological experience) and a neurosurgeon (S.W.L., 16 years of neurosurgical experience) by comparing post-treatment brain MRI findings. In equivocal cases, a consensus was reached. To evaluate the effectiveness of first-line chemotherapy, a complete response (CR) was characterized by the complete elimination of all contrast-enhancing tumors on follow-up MRI scans. A partial response (PR) was characterized by a reduction of at least 50% in the volume of contrast-enhancing tumors [20]. Stable disease (SD) referred to a condition that did not meet the criteria for PR but also did not indicate progressive disease (PD). PD was identified by an increase of more than 25% in the volume of contrast-enhancing tumors, the emergence of new enhancing tumors on follow-up MRI, or involvement of the eye or cerebrospinal fluid [19,20]. Follow-up brain MRIs were conducted every 3 to 6 months during the treatments. For evaluating isolated CNS or systemic tumor relapses, contrast-enhanced CT/MRI and PET scans, as well as clinical or laboratory findings, were utilized, following guidelines from an international workshop aimed at standardizing baseline evaluations and response criteria for PCNSL. CR, PR, and SD were categorized as non-R/R diseases, while PD fell under R/R disease. Progression-free survival (PFS) was measured from the start of effective treatment to the point when tumor progression, recurrence, or death occurred during the follow-up period.

### 3.5. Imaging Acquisition

Preoperative brain MRI scans were performed using either a 1.5-T scanner (Siemens Avanto, Erlangen, Germany; Siemens Aera, Erlangen, Germany; or GE Signa, Milwaukee, WI, USA) in 42 cases or a 3-T scanner (GE Discovery MR750, Boston, MA, USA) in 4 cases. All scanners were equipped with eight-channel head coils. The MRI protocols utilized a range of imaging techniques to provide comprehensive evaluations. These included spin-echo T1-weighted imaging (T1WI) for assessing anatomical structures and tissue characteristics, fast spin-echo T2-weighted imaging (T2WI) for identifying changes in tissue water content and pathology, and T2-fluid attenuated inversion recovery (FLAIR) imaging to detect lesions and abnormalities in areas with high fluid content. Additionally, diffusion-weighted imaging (DWI) was employed to evaluate the movement of water molecules within tissues, while ADC mapping was used to quantify this diffusion. Contrast-enhanced T1-weighted imaging (CE-T1WI) was also incorporated to enhance visualization of contrast-accumulating lesions and vascular structures. DWI was performed with b-values of 1000 and 1500 s/mm^2^, capturing images sequentially in the x, y, and z axes from which ADC maps were derived. The contrast-enhanced images in both axial and coronal T1WI were obtained following the intravenous injection of gadobutrol (Gadovist) or gadoterate meglumine (Dotarem) at a dosage of 0.1 mmol/kg of body weight. For a comprehensive overview of the MR imaging protocols, please refer to Appendix A.

### 3.6. Tumor Segmentation and Quantitative Feature Extraction

Deep learning models, particularly those involving neural networks like nnU-Nets, have indeed redefined the benchmarks in medical image segmentation with their robust performance across a variety of tasks. Nonetheless, the effectiveness of these advanced models typically hinges on the availability of large, well-annotated datasets. Our study operates within the constraints of a relatively small dataset specific to PCNSL, which poses unique challenges not only in terms of volume but also regarding the specific characteristics of the lesions involved. Given these constraints, our design choice was guided by the need for a reproducible, robust, and easily verifiable segmentation method. Therefore, we opted for a semi-automatic tool based on k-means clustering, which, while simpler, provides a level of consistency and reliability particularly suited to smaller datasets. This approach ensures that the segmentation quality remains uniform across this study, minimizing the variability that can arise from more complex deep learning models when trained on limited data.

For each patient, the tumor region of interest (ROI) was delineated on CE-T1WI using the fuzzy C-means (FCM) clustering algorithm by an experienced neuroradiologist (C.C.K.) (Figure 1) [21]. The neuroradiologist determined the tumor location and slice range, and the rectangular box contained all the lesion regions. The lesion ROI was automatically segmented on each slice and combined to a volume of interest (VOI) using FCM based on 3D connected-component labeling. To ensure that VOI corresponded to all MRI sequences, the segmented VOI was co-registered to T2WI and FLAIR by using the open-source ITK-SNAP 4.0 [22]. Finally, the registered MRI images and VOIs were converted to a nearly raw raster data (NRRD) format. Radiomic features were extracted from CE-T1WI, T2WI, and FLAIR images using PyRadiomics 3.0 in Python [23]. By focusing on the segmented tumor regions, we ensured that the features were computed only from the tumor lesion, thereby avoiding any influence from non-tumor tissues. A total of 107 radiomic features were extracted for each MR sequence, including 14 shape-based features, 18 first-order statistical features, and 75 texture features. Shape-based features were used as a VOI lesion mask to analyze the shape of the lesion region. First-order statistical features were employed to characterize the distribution of voxel intensities within the VOIs, while texture features were utilized to quantify the heterogeneity of voxels within these regions. After extracting the features, each one was normalized using the Z-score method. This process involved subtracting the mean value of the feature and then dividing the result by the standard deviation of the feature. For feature selection, the support vector machine (SVM) algorithm obtains optimal feature subsets to minimize the misclassification rate. To evaluate the importance of the extracted features for predicting R/R in PCNSL, a SVM with a Gaussian kernel method was used to select the five most important features [24]. Various ML algorithms have been used to build radiomic models with 10-fold cross-validation, including SVM, k-nearest neighbors (KNN), linear discriminant analysis (LDA), naïve bayes (NB), and decision trees (DT) algorithms. Moreover, the radiomics score was calculated for each patient to predict the R/R probability in PCNSL using a threshold of 0.5. Figure 1 shows a flowchart for establishing the radiomics-based predictive models. The ML models in our research were implemented using MATLAB’s Classification Learner app, adhering strictly to default settings. This decision was informed by initial testing phases that suggested these settings provided a stable baseline for model performance. Using default parameters also enhances the reproducibility of our results, as it simplifies the process for other researchers to validate and build upon our findings without extensive custom configuration, which is particularly advantageous given our limited dataset size. For feature extraction in our study, we utilized MATLAB’s sequential function integrated with an SVM using a radial basis function kernel. This method allowed us to selectively identify and retain the most predictive features through a forward selection process. We conducted this selection using a 10-fold cross-validation setup, optimizing for accuracy and ensuring robustness by performing 500 Monte Carlo repetitions to effectively estimate generalization error. This approach was chosen for its proven capability in handling non-linear relationships within our high-dimensional data, thereby ensuring the reproducibility and reliability of our feature selection process.

In this study, the dataset comprising MRI scans of PCNSL was divided using a random stratification approach. This method ensured that each subset of the dataset—both training and testing—accurately reflects the overall distribution of key clinical variables and outcomes within the full dataset. The random split is particularly useful in avoiding biases that could skew the model’s learning process, ensuring a fair assessment of its predictive capabilities. Furthermore, our study employed a 10-fold cross-validation strategy, which is widely regarded as a robust approach for model evaluation, particularly in scenarios where the utmost generalizability of the findings is crucial. By dividing the data into ten subsets and iteratively using one subset for testing while the others are used for training, we can mitigate the risks of overfitting and underfitting—common concerns in ML studies. The results of this cross-validation process provided us with consistent performance metrics, reinforcing our confidence in the model’s ability to generalize beyond our specific dataset to other clinical settings. 

### 3.7. Measurement of Apparent Diffusion Coefficient Value

To evaluate the performance of ML models in predicting R/R of PCNSL, ADC values (b = 1000 s/mm^2^) from DWI were manually assessed by two skilled neuroradiologists (C.C.K. and S.W.L.). A circular region of interest (ROI), with an area between 22 and 64 mm^2^ (average 36 mm^2^), was strategically positioned within a solid enhancing tumor region to minimize the impact of surrounding necrotic, hemorrhagic, or cystic areas on the ADC measurements (Figure 2). Due to the high level of agreement between the two raters, the statistical analysis of the ADC values utilized the average of the measurements obtained by both raters.

### 3.8. Statistical Analysis

Statistical evaluations were conducted utilizing SPSS software (version 24.0; IBM, Chicago, IL, USA). To analyze categorical data, the chi-square test or Fisher’s exact test was employed, while the Mann-Whitney U test was applied for continuous data. Receiver operating characteristic (ROC) analyses were carried out for the SVM, KNN, LDA, NB, and DT algorithms to differentiate between patients with R/R disease and those without. The Kaplan–Meier method was used to evaluate PFS, and the log-rank test assessed the significance of the R/R rates. A *p* value of less than 0.05 was deemed statistically significant.

## 4. Results

### 4.1. Clinical Information and MRI Results

Table 2 provides a summary of the clinical information and MRI findings for individuals with PCNSL, comparing those with and without R/R. Of the 46 PCNSL patients included in this study, 20 (20/46, 43.5%) patients were found to have R/R (Figure 2), and 26 (26/46, 56.5%) patients remained non-R/R. Out of 46 patients, 27 (58.7%) achieved a CR, 12 (26.1%) had a PR or SD, and 7 (15.2%) exhibited PD following first-line chemotherapy. The objective response rate was 84.8% (39/46), and 20 patients (43.5%) passed away. The median scores for SVM, KNN, LDA, NB, and DT were higher in the R/R group compared to the non-R/R group. However, the median ADC values were significantly higher in the non-R/R group (*p* < 0.05) (Figure 3). Mortality rate was lower in the non-R/R group compared to those with relapse (*p* < 0.05). The overall median follow-up period for all participants was 27.2 months, while the 20 patients who experienced R/R had a median time to relapse of 13 months.

### 4.2. Performance of ML Algorithms and Survival Analysis

The most significant five radiomic features (GLSZM large area low gray level emphasis, first-order kurtosis, original first-order minimum, original GLCM cluster tendency, original GLRLM gray level non-uniformity normalized) selected by the SVM classifier were used to establish five predictive models for differentiation of R/R in PCNSL. The performances of the different models in the validation set are summarized in Table 3. Among the prediction models, the SVM demonstrated the best performance for predicting R/R in PCNSL, with an overall prediction accuracy of 83%, precision of 80%, and AUC of 0.78. The ROC analyses of all five ML models and ADC values are shown in Figure 4. When comparing the tumor progression trends, patients with higher SVM and NB scores (higher than the cut-off points) exhibited significantly shorter PFS (*p* < 0.05) (Figure 5).

## 5. Discussion

This study attempted to use pretreatment brain MRI-based radiomics analysis to predict the clinical outcomes of PCNSL. Based on pretreatment MRI including CE T1WI, T2WI, and FLAIR, SMV offered the best performance for predicting R/R PCNSL among the different ML algorithms. Furthermore, the SVM model outperformed the manually measured ADC values in predicting relapse or recurrence in PCNSL. In terms of tumor progression, patients who had elevated SVM and NB scores demonstrated a notably reduced PFS.

PCNSL is a highly aggressive form of cancer, with its incidence on the rise among individuals aged 65 and older [25]. While some patients initially show positive responses to chemotherapy and radiation therapy, approximately 50–60% ultimately experience relapsed or refractory disease. The best treatment approach for PCNSL is still a matter of debate. On standard MRI scans, PCNSL typically presents as a single enhancing mass located in the supratentorial region, often affecting areas such as the basal ganglia, corpus callosum, fornix, and periventricular regions [26].

The current standard response criteria for lymphoma are the revised Cheson criteria [20] and Lugano criteria [27]. These criteria assess treatment outcomes by comparing 18F-FDG PET and CT images taken before and after therapy. To date, only a limited number of imaging biomarkers have been identified for predicting clinical outcomes in PCNSL. A recent study by Krebs et al. [28] highlighted that elevated 18F-FDG uptake and an increased volume of 18F-FDG-avid lesions correlate with unfavorable prognoses in PCNSL patients. Currently, DWI is extensively utilized as a biomarker in oncology, applicable to both intracranial and extracranial tumors. Additionally, DWI serves as a tool for assessing tumor cell differentiation. This imaging modality provides critical biomedical information by analyzing the random Brownian motion of water molecules within tissue voxels, which is quantitatively represented by the ADC values. Consequently, atypical ADC results may indicate underlying abnormalities in biological tissues [29]. Chien et al. [9] and Baek et al. [30] reported that lower pretreatment ADC values were associated with higher recurrence rates and shorter PFS in PCNSL. However, the ADC values were measured by manually placing the ROI, and inter-rater inconsistencies may have occurred when measuring the data.

Radiomics is a novel imaging method used to investigate the association among cancer imaging, genotyping, and clinical outcomes in the era of precision medicine. Although radiomic analyses for distinguishing PCNSL from brain glioblastoma and metastasis have been reported in several studies [12,13], the radiomic-based models for the prediction of clinical outcomes in PCNSL have only been reported in rare studies with small sample sizes. Recently, Destito et al. [15] used ML models to predict the OS and PFS in 23 patients with PCNSL. The results showed that the radiomics-based prediction model is superior to the clinical factor-based model, with AUC of 0.86 and 0.84 for the prediction of OS and PFS, respectively. Ali et al. [14] used CE T1 and T2-FLAIR radiomic features in the SVM network for predicting treatment response to MTX-based induction in 47 PCNSL patients; an accuracy of 81.1% and AUC of 0.81 were obtained. Chen et al. [18] also reported that textural features of CE T1WI could potentially serve as prognostic biomarkers in 52 patients with PCNSL. She et al. [17] first attempted to use deep learning to predict OS in 56 PCNSL patients, and AUC of 0.81, accuracy of 87%, and precision of 88% were obtained in CE T1WI. These results suggest that ML-based radiomics analysis can predict tumor aggressiveness in PCNSL and has the potential to be integrated into clinical predictive tools. The present study first compared ML algorithms and ADC for the prediction of clinical outcomes in PCNSL, and higher AUC in SVM, KNN, NB, and DT predictive models were observed as compared with the ADC-based model.

According to the features selected from our dataset, which were mostly high-dimensional, SVM with a Gaussian kernel outperformed the others. This is crucial given the complexity of image feature space. NB also performed well, suggesting that a probabilistic approach is sufficient for the level of image feature interdependencies present. The KNN and DT provided moderate accuracy. The lower performance of LDA suggests that its linear assumptions did not align well with the structure of our dataset, leading to the weakest performance. These results underscore the importance of the algorithm selection in the ML method, where data characteristics can affect the model performance. In clinical practice, Kotowski et al. [31] reported that clinically inspired radiomic features significantly improve the performance of ML in detecting liver cirrhosis in computed tomography (CT). However, MRI differs fundamentally from CT in terms of the image features and physical properties, and no corresponding imaging study in PCNSL has been reported. One of the major challenges in clinical practice is the lack of reproducibility and generalizability of the reported radiomic features and models. The reproducibility or repeatability of radiomic features may be related to intra-individual test-retest differences, image machines, image acquisition, and reconstruction parameters [32]. For PCNSL, Destito et al. [15] found that features derived from Z-score normalized images exhibited significantly greater stability compared to those obtained from non-normalized images, showing an average enhancement of 38%. Nenning et al. [16] used CE T1WI textural features obtained from 191 patients with PCNSL across nine sites in Austria and reported that the radiomic risk score is a robust and complementary predictor of survival and is reflected at the level of DNA methylation in PCNSL. Shiri et al. [33] reported that up to 74% of MRI texture features of glioblastomas had high reproducibility. In contrast, Mitchell-Hay et al. [34] reported that the repeatability and reproducibility of variables are significant limitations of radiomic analysis in 3D T1W brain MRI and that careful selection of radiomic features is required. In a phantom study conducted by Baeßler et al. [35], it was found that merely one-third of the assessed features demonstrated strong robustness and reproducibility across various MRI sequences. Nevertheless, the study suggests that the influence of operator-dependent bias on radiomic features can be minimized through the use of fully automated image segmentation techniques.

Compared with previous research, which mostly used CE TIWI as input, our study integrated CE T1WI, T2WI, and T2 FLAIR images and compared the results in different ML- and ADC-based models. However, this study had several limitations. First, selection bias may exist owing to the retrospective nature of this study and the lack of external validation. The images were acquired from two medical institutions, and further testing using multi-institutional data with different MRI protocols is necessary to determine whether the trained predictive classifier is generalizable. Variations in MRI scanners and magnetic field strengths may have influenced the MRI characteristics observed. Additionally, due to the limited sample size, we opted for ML techniques to generate predictions while minimizing the risk of overfitting.

## 6. Conclusions

Our initial research indicated that ML utilizing preoperative MR radiomic features could serve as an effective method for predicting relapse or recurrence in patients with PCNSL. The imaging characteristics obtained through automated segmentation and image registration were both quantitative and objective. These results offer valuable information for developing treatment strategies in PCNSL. They assist in choosing appropriate chemotherapy drugs and guide decisions regarding supplementary consolidation therapies, including radiation therapy or stem cell transplantation. As more cases are collected in the future, the implementation of deep learning models, like convolutional neural networks, could enhance predictive accuracy.

## Figures and Tables

**Figure 1 life-14-01290-f001:**
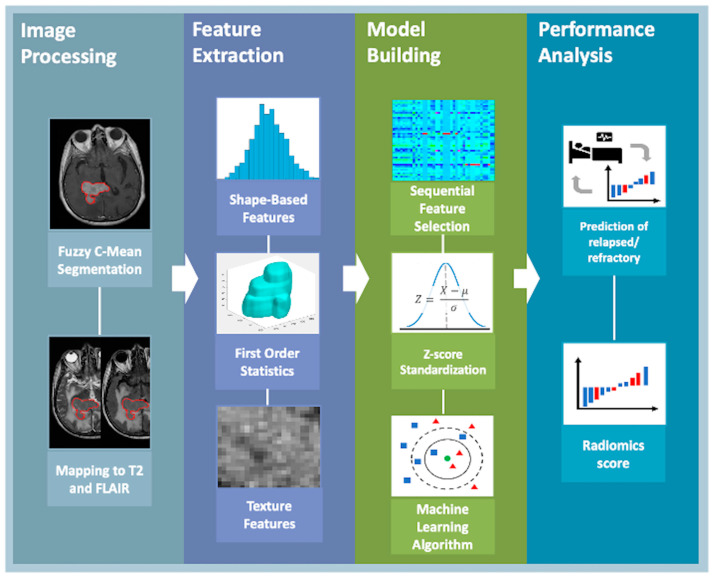
Flowchart for building the radiomics-based predictive model. The PCNSL is segmented by a fuzzy c-means clustering algorithm on contrast-enhanced T1WI, and the segmented ROI is mapped to T2WI and T2 FLAIR. In feature extraction, a total of 107 radiomic features, including 14 shape-based features, 18 first-order statistics features, and 75 texture features in each imaging sequence, were extracted. Further, the most important 5 features were selected by SVM, and each feature was normalized by the Z-score method. Subsequently, predictive models were built using five different ML algorithms to predict R/R PCNSL.

**Figure 2 life-14-01290-f002:**
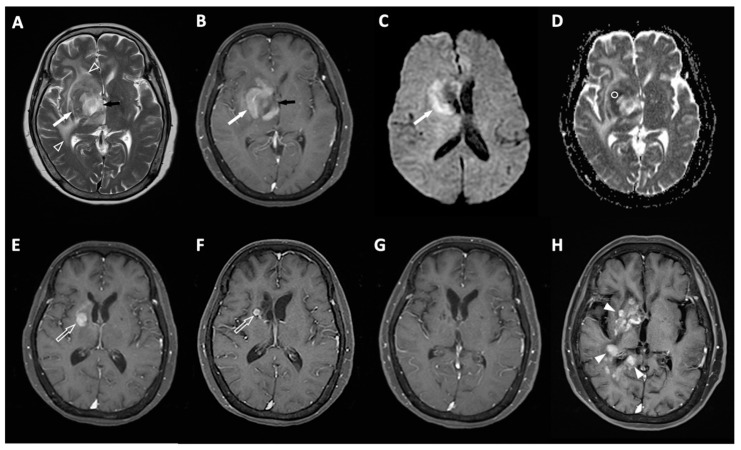
A 74-year-old woman was diagnosed with PCNSL pathologically. Imaging studies included (**A**) axial T2WI and (**B**) axial contrast-enhanced T1WI, which identified an enhancing tumor (white arrow) in the right basal ganglia, along with peritumoral edema (open arrowhead) and intratumoral necrosis (black arrow). (**C**) DWI showed hyperintensity in the tumor (white arrow), suggesting restricted random motion of water molecules. (**D**) The ADC value, measured within a defined circular region, was 0.56 × 10^−3^ mm^2^/s. In ML algorithms, the computed scores were as follows: 1.12 for SVM, 0.78 for KNN, 0.69 for LDA, 0.89 for NB, and 0.77 for DT. (**E**–**G**) Following first-line chemotherapy, a reduction in tumor size (open arrow) was noted, leading to a complete response (**G**). (**H**) However, 51 months later, recurrent tumors (arrowheads) were detected.

**Figure 3 life-14-01290-f003:**
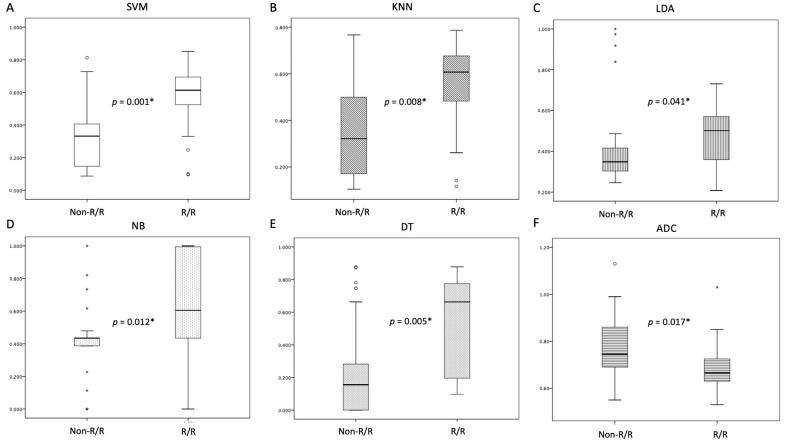
The box plots illustrate the values for (**A**) SVM, (**B**) KNN, (**C**) LDA, (**D**) NB, (**E**) DT, and (**F**) ADC in PCNSL patients with and without R/R disease. The R/R group exhibited higher scores for SVM, KNN, LDA, NB, and DT, alongside lower ADC values when compared to the non-relapsed group. * Statistical difference (*p* < 0.05). The boxes in the plots represent the interquartile range, while the whiskers extend to indicate the full range of the data. The median value for each category is marked by a horizontal line within the box. Outliers are depicted as circles, which are defined as data points falling more than 1.5 times the interquartile range below the first quartile or above the third quartile. Additionally, extreme values are indicated by stars, representing those that exceed three times the interquartile range above the third quartile.

**Figure 4 life-14-01290-f004:**
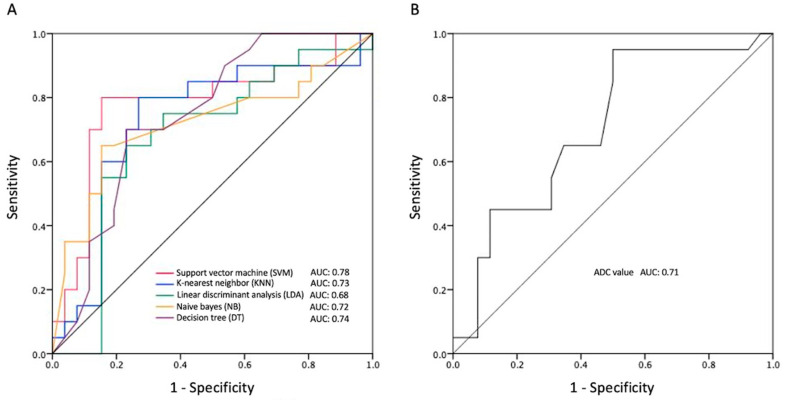
The ROC curves were analyzed for two categories: (**A**) MRI-based radiomic ML algorithms and (**B**) ADC values in predicting R/R PCNSL. The AUC values for the various models were as follows: SVM achieved an AUC of 0.78, followed by KNN at 0.73, LDA at 0.68, NB at 0.72, DT at 0.74, and the ADC model at 0.71.

**Figure 5 life-14-01290-f005:**
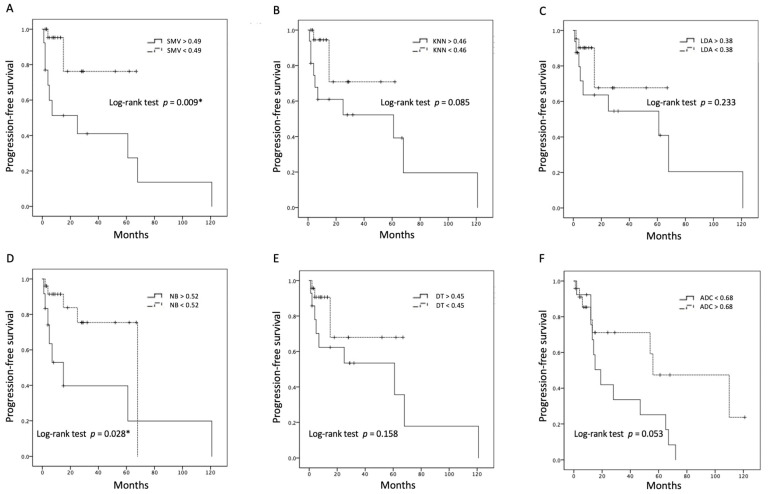
Kaplan–Meier curves illustrating overall progression-free survival trends based on cut-off points for (**A**) SMV, (**B**) KNN, (**C**) LDA, (**D**) NB, (**E**) DT, and (**F**) ADC values. * Statistical difference (*p* < 0.05).

**Table 1 life-14-01290-t001:** The strengths and weaknesses of relevant papers on the use of radiomics to predict PCNSL prognosis.

Reference	Specific Methods	Strengths	Weakness
Destito et al. [15]	Extra Tree Classifier SVM Logistic regression Random forest k-neighbors	Novel ML techniques, potentially improving predictive accuracy for survival rates compared to traditional statistical methods.	Potential limitations in generalizability due to image extraction only from T1WI and T2WI.
Nenning et al. [16]	Least absolute shrinkage and selection operator (LASSO) SVM	Integration of multi-modal data including DNA and MRI for radiomic score in predicting survival.	Only single contrast-enhanced T1WI for feature extraction. Interpreting complex texture-based radiomic features remains challenging for clinical application.
She et al. [17]	3D-ResNet-based deep learning (DL) SVM	Implements visual explanations of the model’s predictions.	Deep learning models heavily rely on the quantity and quality of the data used for training.
Ali et al. [14]	SVM Multinomial naive bayes	Incorporating radiomic features with clinical data enhances prediction accuracy for treatment response and survival outcomes.	Limited details on specific feature selection reduce transparency and model interpretability.
Chen et al. [18]	Multivariate analyses Cox proportional hazards regression	Texture analysis offers a quantitative method for assessing tumor heterogeneity and predicting survival.	Limited biological interpretation of image features as independent survival predictors.

**Table 2 life-14-01290-t002:** Clinical data and MRI results in individuals with PCNSL.

	Relapsed/Refractory (R/R)	Non-R/R	*p* Value
**Number of patients**	20	26	
**Sex**			0.139
Male	6 (30%)	14 (53.8%)	
Female	14 (70%)	12 (46.2%)	
**Age (y)**	62 (54.5, 69.5)	66.5 (57, 76)	0.287
**Response to first-line chemotherapy**			0.069
Complete response (CR)	8 (40%)	19 (73.1%)	
Partial response (PR)/Stable disease (SD)	7 (35%)	5 (19.2%)	
Progressive disease (PD)	5 (25%)	2 (7.7%)	
**Tumor location**			0.449
Cerebral cortex	11 (55%)	18 (69.2%)	
Basal ganglia/thalamus/corpus callosum	7 (35%)	5 (19.2%)	
Cerebellum	2 (10%)	3 (11.5%)	
**Ocular involvement**	3 (15%)	2 (7.7%)	0.640
**Enhancement**			0.855
Homogeneous	9 (45%)	11 (42.3%)	
Heterogeneous	11 (55%)	15 (57.7%)	
**Necrosis**	7 (35%)	10 (38.5%)	0.809
**Hemorrhagic change**	4 (20%)	8 (30.8%)	0.467
**Peritumoral edema**	19 (95%)	24 (92.3%)	1
**Leptomeningeal seeding**	3 (15%)	2 (7.7%)	0.640
**Multiple lesions**	11 (55%)	11 (42.3%)	0.393
**Maximal tumor diameter (cm)**	3.9 (3.1, 4.8)	3.2 (2.3, 4.2)	0.245
**High DWI signal**	18 (90%)	21 (80.8%)	0.446
**ADC value (×10^−3^ mm^2^/s)**	0.67 (0.62, 0.72)	0.75 (0.66, 0.83)	0.017 *
**SVM score**	0.61 (0.52, 0.71)	0.33 (0.20, 0.47)	0.001 *
**KNN score**	0.61 (0.51, 0.71)	0.32 (0.15, 0.49)	0.008 *
**LDA score**	0.50 (0.39, 0.62)	0.35 (0.28, 0.42)	0.041 *
**NB score**	0.61 (0.32, 0.89)	0.44 (0.39, 0.48)	0.010 *
**DT score**	0.66 (0.37, 0.96)	0.16 (0.13, 0.34)	0.005 *
**LDH (units/L)**	204 (181, 227)	225 (158, 292)	0.194
**Ki-67 (%)**	85 (79, 92)	85 (76, 95)	0.758
**Recurrence site**			
CNS	19 (95%)		
Isolated systemic	0		
Both CNS and systemic	1 (5%)		
**Death**	12 (60%)	8 (30.8%)	0.047 *
**Follow-up time (months)**	31.1 (17, 45.2)	22.6 (6.1, 39.1)	0.198

* Statistical difference (*p* < 0.05). Continuous variables were reported using the median along with the interquartile range.

**Table 3 life-14-01290-t003:** Performance of ML algorithms and ADC for differentiating PCNSL with R/R.

	Accuracy	Precision	AUC	Cut-Off Value	*p* Value
**SVM score**	0.83	0.80	0.78 (0.63, 0.93)	0.49	0.001 *
**KNN score**	0.74	0.70	0.73 (0.57, 0.89)	0.46	0.008 *
**LDA score**	0.72	0.73	0.68 (0.51, 0.84)	0.38	0.041 *
**NB score**	0.76	0.76	0.72 (0.56, 0.88)	0.52	0.012 *
**DT score**	0.74	0.70	0.74 (0.60, 0.89)	0.45	0.005 *
**ADC value**	0.78	0.86	0.71 (0.55, 0.86)	0.68	0.017 *

* Statistical difference (*p* < 0.05).

## Data Availability

Data available on request due to privacy and ethical restrictions.

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
