# Peer review of "MRI-Based Machine Learning for Prediction of Clinical Outcomes in Primary Central Nervous System Lymphoma"

_life, 2024, doi:10.3390/life14101290_

Round 1
Reviewer 1 Report
Comments and Suggestions for Authors
The sections of the proposed paper should be improved as follows.
- Please insert in the “Introduction” contribution and the last paragraphs contain a short description of the paper structure.
- In this study, the authors used a “total of 107 radiomic features, including 14 shape-based, 18 first-order statistical, and 75 texture features." Unfortunately, it is not specified if these were computed from the whole image because the used features are sensitive if they are not extracted from the region of interest, in this case, only cerebral matter. An example of an extracted region of interest (with skull stripping segmentation method) in the following study can be found out.https://www.ncbi.nlm.nih.gov/pmc/articles/PMC4636724/
- A “Related Work” section is necessary; the authors will add novelty studies that use approximately the same subject, machine learning classifiers, or other AI tools. This section should also include the weaknesses and strengths of state-of-the-art methods.
- The authors used for classification the “support vector machine (SVM), k-nearest neighbors (KNN), linear discriminant analysis (LDA), naïve Bayes (NB), and decision trees (DT). Please validate the results with a method such as bagging, boosting, or stacking. An example in the valuable papers can be found (https://www.mdpi.com/2313-433X/10/1/8 and https://ieeexplore.ieee.org/document/10566200).
- In the section "4. Discussion," please insert the comparison with state-of-the-art methods as a table; the actual form is hard to follow.
- The quality of Figures 3 and 5 should be improved; on the axis, the number values are intelligible.
- Please add the limitations of the proposed study.
- In the “Conclusion” section, please add the future research directions.
Author Response
Response to reviewer 1
Comments 1:
Please insert in the “Introduction” contribution and the last paragraphs contain a short description of the paper structure.
Response 1:
Thanks to reviewer’s comment We make revision according to suggestion ( lines 66-68).
Comments 2:
In this study, the authors used a “total of 107 radiomic features, including 14 shape-based, 18 first-order statistical, and 75 texture features." Unfortunately, it is not specified if these were computed from the whole image because the used features are sensitive if they are not extracted from the region of interest, in this case, only cerebral matter. An example of an extracted region of interest (with skull stripping segmentation method) in the following study can be found out. https://www.ncbi.nlm.nih.gov/pmc/articles/PMC4636724/
Response 2:
Thanks to reviewer’s comment. We confirmed that the 107 radiomic features were computed from the extracted region of interest (ROI) of the segmented tumor, not from the whole image. Although we did not perform skull stripping, the combination of precise tumor segmentation and co-registration across MRI sequences ensured that the features were extracted solely from the tumor ROI. Please refer to the section of “Tumor Segmentation and Quantitative Feature Extraction” (lines 193-204). Some revision was made (marked in red).
Comments 3:
A “Related Work” section is necessary; the authors will add novelty studies that use approximately the same subject, machine learning classifiers, or other AI tools. This section should also include the weaknesses and strengths of state-of-the-art methods.
Response 3:
Thanks to reviewer’s comment. We added “Related Work” section as suggestion (lines 70-104). Although the weaknesses and strengths of methods cannot be evaluated definitely in small data studies, we added a table (Table 1) for additional explanation in these studies.
Comments 4:
The authors used for classification the “support vector machine (SVM), k-nearest neighbors (KNN), linear discriminant analysis (LDA), naïve Bayes (NB), and decision trees (DT). Please validate the results with a method such as bagging, boosting, or stacking. An example in the valuable papers can be found (https://www.mdpi.com/2313-433X/10/1/8 and https://ieeexplore.ieee.org/document/10566200).
Response 4:
Thanks to reviewer’s comment. We appreciate your recommendation to validate our results using ensemble learning methods such as bagging, boosting, or stacking. These techniques have indeed shown effectiveness in improving classification performance, particularly when large datasets are available. In our study, we are working with a relatively small dataset. Ensemble methods like bagging, boosting, and stacking typically require a substantial amount of training data to build diverse and robust models. As indicated in the literature, these methods can effectively leverage the variability in the data to improve predictive accuracy when thousands of training samples are available. However, with a limited dataset size, as in our case, applying these techniques may not provide significant benefits and could potentially lead to overfitting.
Therefore, in the current study, we focused on individual classifiers such as SVM, KNN, LDA, NB, and DT, which are common methods in machine learning and have been successfully used in similar contexts. We are continuously collecting data from such patients, and we plan to apply ensemble learning methods in future research when a larger dataset becomes available. We believe that with more extensive data, ensemble techniques could significantly enhance the predictive performance of our models.
Comments 5:
In the section "4. Discussion," please insert the comparison with state-of-the-art methods as a table; the actual form is hard to follow.
Response 5:
Thanks to reviewer’s comment. We summarized state-of-the-art methods in Table 3 ( lines 344-346).
Comments 6:
The quality of Figures 3 and 5 should be improved; on the axis, the number values are intelligible.
Response 6:
Thanks to reviewer’s comment. We replaced the original figures with high-quality ones.
Comments 7:
Please add the limitations of the proposed study.
Response 7:
Thanks to reviewer’s comment. We mentioned limitations in the section of discussion (lines 441-458).
Comments 8:
In the “Conclusion” section, please add the future research directions.
Response 8:
Thanks to reviewer’s comment. We make revision according to suggestion ( lines 456-458).

Reviewer 2 Report
Comments and Suggestions for Authors
In this paper, the authors tackled an important problem of automated prediction of clinical outcomes in primary central nervous system lymphoma. The topic is certainly worthy of investigation, and it easily falls into the scope of the journal. There are, however, quite a number of shortcomings which need to be, in my opinion, thoroughly addressed before the manuscript could be considered for publication:
1. The authors should make sure that each abbreviations is defined at its fist use. Also, the abstract is unnecessarily packed with lots of abbreviations – I suggest moving them to the main body of the manuscript to enhance the read.
2. I strongly encourage the authors to revisit the current state of the art in the context of applying radiomic features to analyzing medical image data. There have been lots of techniques proposed so far in this vein, also expanding feature extractors (that are radiomic) with those that correspond to clinically-relevant features (See e.g., the work by Kotowski: https://pubmed.ncbi.nlm.nih.gov/36512877/). Would such an approach be useful in the context of lymphoma? Please discuss.
3. There are numerous well established techniques for delineating lesions in medical image data, such as nnU-Nets by Isensee which actually established the state of the art in the field. The authors should discuss their design choices in more detail.
4. We are currently facing the reproducibility crisis in the machine learning field. To tackle this, please provide a link to the repository containing the implementation of the method, as reimplementing it would certainly not be trivial.
5. The quality of the figures should be improved – all of them should be high-resolution in a vector format.
6. The authors should thoroughly discuss the training-test dataset split used in this study. Also, I strongly recommend following multi-fold cross-validation to ensure that the insights learned from this study could generalize.
7. The authors should thoroughly discuss all hyperparameters used in this study (and concerning the machine learning models, as well as exploited feature extractors).
8. The authors did not provide any qualitative (visual) analysis – are there any specific thus interesting patients for which the system “failed”?
9. While reporting AUC, please also report confidence intervals.
10. Although the English is acceptable overall, the manuscript would still greatly benefit from careful proofreading – I spotted several grammatical errors around the manuscript.
Comments on the Quality of English LanguageThe manuscript would benefit from proofreading.
Author Response
Response to reviewer 2
Comments 1:
The authors should make sure that each abbreviation is defined at its first use. Also, the abstract is unnecessarily packed with lots of abbreviations – I suggest moving them to the main body of the manuscript to enhance the read.
Response 1:
Thanks to reviewer’s comment. We make revision according to suggestion (lines 21-39).
Comments 2:
I strongly encourage the authors to revisit the current state of the art in the context of applying radiomic features to analyzing medical image data. There have been lots of techniques proposed so far in this vein, also expanding feature extractors (that are radiomic) with those that correspond to clinically-relevant features (See e.g., the work by Kotowski: https://pubmed.ncbi.nlm.nih.gov/36512877/). Would such an approach be useful in the context of lymphoma? Please discuss.
Response 2:
Thanks to reviewer’s comment. While Kotowski et al.’s approach has demonstrated significant advancements in feature extraction and dimensionality reduction within computed tomography scans for liver cirrhosis, direct application to MRI-based studies, such as ours focusing on PCNSL, requires careful consideration. MRI differs fundamentally from CT in terms of the image features physical properties they capture. However, the methodology of reducing feature dimensionality while retaining clinically relevant information could be insightful for enhancing MRI-based predictive models. When more cases were collected in the future which would allow for rigorous external validation, we are interested in exploring how such dimensionality reduction methods could be adapted to the radiomic features derived from MRI scans.
Revision:
Additional explanation of this issue was added in discussion (lines 416-420).
Comments 3:
There are numerous well established techniques for delineating lesions in medical image data, such as nnU-Nets by Isensee which actually established the state of the art in the field. The authors should discuss their design choices in more detail.
Response 3:
Thanks to reviewer’s comment. We added additional instructions as suggestion.
Revision:
Additional explanation of this issue was added in the section of “ Tumor segmentation and quantitative feature extraction” (lines 180-192).
Comments 4:
We are currently facing the reproducibility crisis in the machine learning field. To tackle this, please provide a link to the repository containing the implementation of the method, as reimplementing it would certainly not be trivial.
Response 4:
Thanks to reviewer’s comment. We provided references for each software package used in the analysis process in this article. Addressing the crucial issue of reproducibility in machine learning, particularly in medical imaging, we recognize the importance of making our methodologies accessible and executable. We will consider publishing these resources in the future to facilitate easy access and enhance the reproducibility of our research.
Comments 5:
The quality of the figures should be improved – all of them should be high-resolution in a vector format.
Response 5:
Thanks to reviewer’s comment. We replaced the original figures with high-quality ones in a vector format.
Comments 6:
The authors should thoroughly discuss the training-test dataset split used in this study. Also, I strongly recommend following multi-fold cross-validation to ensure that the insights learned from this study could generalize.
Response 6:
Thanks to reviewer’s comment. We added additional instructions as suggestion.
Revision:
Additional explanation of this issue was added in the section of “ Tumor segmentation and quantitative feature extraction” (lines 236-248).
Comments 7:
The authors should thoroughly discuss all hyperparameters used in this study (and concerning the machine learning models, as well as exploited feature extractors).
Response 7:
Thanks to reviewer’s comment. We added additional instructions as suggestion.
Revision:
Additional explanation of this issue was added in the section of “ Tumor segmentation and quantitative feature extraction” (lines 221-235).
Comments 8:
The authors did not provide any qualitative (visual) analysis – are there any specific thus interesting patients for which the system “failed”?
Response 8:
Thanks to reviewer’s comment. We summarized the qualitative (visual) analysis in the Table 2, and most of them did not show significant difference between recurrent and non-recurrent groups.
Comments 9:
While reporting AUC, please also report confidence intervals.
Response 9:
Thanks to reviewer’s comment. We make revision according to suggestion (Table 3, marked in red).
Comments 10:
Although the English is acceptable overall, the manuscript would still greatly benefit from careful proofreading – I spotted several grammatical errors around the manuscript.
Response 10:
Thanks to reviewer’s comment. We make careful proofreading thorough the article according to suggestion.

Round 2
Reviewer 2 Report
Comments and Suggestions for Authors
Thank you for addressing most of my concerns.